# Cultivation of Bovine Mesenchymal Stem Cells on Plant-Based Scaffolds in a Macrofluidic Single-Use Bioreactor for Cultured Meat

**DOI:** 10.3390/foods13091361

**Published:** 2024-04-28

**Authors:** Gilad Gome, Benyamin Chak, Shadi Tawil, Dafna Shpatz, Jonathan Giron, Ilan Brajzblat, Chen Weizman, Andrey Grishko, Sharon Schlesinger, Oded Shoseyov

**Affiliations:** 1Department of Plant Sciences, The Robert H. Smith Faculty of Agriculture, Food and Environment, The Hebrew University of Jerusalem, Rehovot 7610001, Israel; 2Sammy Ofer School of Communication, Reichman University, Herzliya 4610101, Israel; jonathan.giron@runi.ac.il (J.G.); ilan.brajzblat@mail.huji.ac.il (I.B.); chen.weizman@post.runi.ac.il (C.W.); andrey.grishko@milab.idc.ac.il (A.G.); 3Department of Animal Sciences, The Robert H. Smith Faculty of Agriculture, Food and Environment, The Hebrew University of Jerusalem, Rehovot 7610001, Israelshadi.tawil@mail.huji.ac.il (S.T.); dafna.shpatz@mail.huji.ac.il (D.S.)

**Keywords:** fluidics, bioreactors, tissue engineering, cultured meat, scaffolds

## Abstract

Reducing production costs, known as scaling, is a significant obstacle in the advancement of cultivated meat. The cultivation process hinges on several key components, e.g., cells, media, scaffolds, and bioreactors. This study demonstrates an innovative approach, departing from traditional stainless steel or glass bioreactors, by integrating food-grade plant-based scaffolds and thermoplastic film bioreactors. While thermoplastic films are commonly used for constructing fluidic systems, conventional welding methods are cost-prohibitive and lack rapid prototyping capabilities, thus inflating research and development expenses. The developed laser welding technique facilitates contamination-free and leakproof sealing of polyethylene films, enabling the efficient fabrication of macrofluidic systems with various designs and dimensions. By incorporating food-grade plant-based scaffolds, such as rice seeded with bovine mesenchymal stem cells, into these bioreactors, this study demonstrates sterile cell proliferation on scaffolds within macrofluidic systems. This approach not only reduces bioreactor prototyping and construction costs but also addresses the need for scalable solutions in both research and industrial settings. Integrating single-use bioreactors with minimal shear forces and incorporating macro carriers such as puffed rice may further enhance biomass production in a scaled-out model. The use of food-grade plant-based scaffolds aligns with sustainable practices in tissue engineering and cultured-meat production, emphasizing its suitability for diverse applications.

## 1. Introduction

Incorporating natural materials, such as plant-based food-grade scaffolds, into bioreactor design aligns with sustainable practices to use renewable animal-free materials, though cost implications remain challenging [1,2,3,4,5,6]. Current research, influenced by seminal works such as [7,8,9,10,11], emphasizes the potential of minimally processed plant-based food-grade scaffolds and the need for cost-effective and innovative solutions for bioprocessing such as single-use bioreactors [12,13,14]. Puffed rice has a highly porous and airy structure due to the expansion of the rice grains during the puffing process. This porosity can provide a three-dimensional environment for cell adherence, proliferation, and nutrient exchange [14,15]. The gelatinization of starches during the production process can render the surface of puffed rice hydrophilic, and its nutritional composition changes [16,17,18]. This property is beneficial for promoting cell adhesion and spreading, as many cells prefer a hydrophilic environment [19,20,21,22]. Puffed rice can be broken down or processed into various sizes and shapes, allowing the scaffold to be customized according to the specific requirements of the bioprocess. The food industry is interested in “clean labels” scaffolds such as puffed rice, which can be manufactured without additives or harmful substances and thus may provide a clean and inert substrate suitable for cell culture, fit within the regulatory framework of the food industry, and be in line with consumer perception [23]. Puffed rice is relatively inexpensive and it is produced globally in tons at a cost of 0.69–0.86 USD/kg [24,25]. In addition to the need for scaffolds to be perfusable and compatible with cells, different scaffolds may lead to different tissue formation due to their mechanical properties [4]. When considering industrial food applications, these scaffolds should be durable enough to go through the process and be introduced into it with a minimal cost, both for the scaffold fabrication process and its sterilization. Finally, in the context of cultured meat, a scaffold that is compatible with cells and supports proliferation in a bioprocess should be regulated as a food. Choosing scaffolds that are already regulated as foods solves one of these issues. Reliable and cost-effective methods to create tailor-made bioreactor systems are vital for advancing biotechnology, proliferating cells in suspension and cultivating cells on scaffolds. In addition, there is a great need for versatility and the ability to culture cells on unconventional scaffolds [26,27,28,29,30,31,32,33,34,35,36,37,38,39,40] for existing industries such as pharma and for the emerging cultured-meat industry [31,41]. However, the high construction cost of bioreactors hinders scalability [5,6,42], and few commercially available large-scale bioreactors support cell cultivation on food-grade scaffolds [26,31].

In this study, we evaluate the use of laser welding to incorporate nylon bilayers, the main component of the macrofluidic single-use bioreactor (MSUB) construction. Our primary goal is to demonstrate compatibility with cell culture and utility as a rapid prototyping method for large-scale fluidics, similar to microfluidics, only at a large scale [43,44,45,46,47,48]. Significantly, we demonstrate that cells can proliferate and potentially differentiate on scaffolds within macrofluidic systems. Using our approach, we demonstrate a decrease in overall bioreactor prototyping and construction costs and potentially answer the need for scalable bioreactor solutions for research and the industry [49,50,51]. This study addresses this gap by leveraging laser welding to integrate nylon bilayers, potentially enabling affordable single-use bioreactors for laboratory-scale and industrial use [52,53,54,55,56]. Converging affordable single-use bioreactors with minimal shear forces and maximizing cell adherence by adding macro carriers such as puffed rice may increase biomass production in a scaled-out model [26]. Bioreactors can provide a controlled environment to support cell growth and tissue development for cultured-meat production. Hollow fiber bioreactors allow cells to attach to a matrix while enabling nutrient and waste exchange [29]. Perfusion bioreactors can continuously supply nutrients and remove waste, matching the structure and size of the cultured tissue [42]. Rotary wall vessel and wave or rocker bioreactors can also be used in batch culture with the potential for perfusion [57,58]. Current macrofluidic devices are made by connecting pumps, sensors, flexible tubes, filters, and chambers made from sterilizable plastic, glass and stainless steel often designed and fabricated by fabrication machines such ash a computer numerical control machine (CNC) or by a 3D printer to fit a bioprocess [59,60,61,62,63,64,65,66,67]. Our method allows one to design and laser-print tubes and connectors in such a way that maintains the sterility of the device while allowing the introduction of filters, pumps, and sensors without compromising sterility or requiring a sterilization step for the macrofluidic device. In summary, our research showcases the potential of laser fabrication of plastic films into macrofluidics, a novel single-use bioreactor fabrication method. This advancement aims to bridge the gap in bioprocessing infrastructure and foster sustainable growth, innovations in cell cultivation for diverse biotechnological, and is uniquely fitting for cultured meat.

## 2. Materials and Methods

### 2.1. Fabrication of Macrofluidic Devices

#### 2.1.1. Laser Fabrication of Macrofluidic Devices from Thermoplastic Film

A laser cutter machine (VLS 3.6 Universal Laser Systems, Inc., Scottsdale, AZ, USA) equipped with a 60 W CO_2_ laser with a programmable focus was used to cut and weld the film. The thermoplastic film bilayer was a polyamide polyethylene (PAPE) (Plastopil, Israel) as shown in Figure 1a. The film had a thickness of 140 µm and consisted of two layers, each 70 µm thick, layered together with high proximity. To calibrate the laser, a raster shape was designed using Adobe Illustrator and was laser-welded (Figure 1b). The laser was then operated over the thermoplastic film bilayer in a continuous motion, generating heat that welded the two film layers together at a power output of 12–24 watts, at maximal speed, in different power settings. The film was placed in the laser cutter, and the laser was focused onto the film layers at z = 0.2 mm for cutting and 8 mm for welding with power according to the calibration assay (Figure 1c); the same procedure was used to calibrate cutting. The welded and cut film was inspected for defects or imperfections using a microscope (model 70, Magiscope, Sarasota, FL, USA) with a 5× eyepiece and 4× objective (Figure 1d). The maximum load of the welded film was tested using an Instron testing machine (Instron 3345 tester, Norwood, MA, USA) equipped with a 100 N load cell (Figure 1e,f). Under optimal settings, this method allowed for a precise and reliable welding and cutting of the polyethylene–polyamide (PAPE) film bilayer with minimal heat-induced damage to the film. The resulting welds were strong and leak-tight, making them suitable for use in fluidic systems and therefore for cell cultivation.

#### 2.1.2. Fabrication of MSUB

The macrofluidic design was RGB: red for cutting and blue for welding (Figure 2a). The film was formed by the laser according to the design (Figure 2b). Sterilized flexible tubes, tube cap, and filters were introduced to the macrofluidic device and secured with zip ties in a laminar flow hood (Figure 2c). The backboard was 3D-printed from ABS, with 14 mm diameter holes (14 mm width), and 3D-printed pegs (Prusa i3 MK3S+, Prusa, Prague, Czech Republic ABS) were used to secure the welded macrofluidic in place and to support the backboard (Figure 2d). Ambient air from the incubator was pumped through tubes and filters into the MSUB constant airflow, maintaining sterile and viable gas, pH, and temperature conditions in the MSUB (Figure 2e). Unlike a stirred tank bioreactor that has high shear forces, this macrofluidic design is useful for cultivating brittle scaffolds while maintaining overhead gas exchange and allowing manual media exchange. The resulting fluidic device did not undergo any internal sterilization post-fabrication. Previous work demonstrated the design and implementation of a similar macrofluidic device for the cultivation of bacteria with a peristaltic pump, a controlled heater, and real-time optical sensing [68].

#### 2.1.3. Air Pump

To supply cells with oxygen and CO_2_ from the incubator environment, we used a 12 V DC 370 mini air pump (Pinya Technologies, Shenzen, China), and an ESP32 microcontroller (Espressif Systems, Shanghai, China) was used to interface with the H-bridge motor driver (Figure 3a). Using its GPIO (General Purpose Input/Output) pins allows control over the air pump direction and speed of the air pump (Figure 3b,c). Power supply connections were established for both the ESP32 and L298N H-bridge, ensuring proper grounding (Figure 3e). Additionally, a user interface accessible via WiFi was implemented on the ESP32, providing a wireless control mechanism (Figure 3d). We developed a program utilizing pulse width modulation (PWM) to achieve precise control over the air pump. Upon successful code compilation and upload, the integrated system allowed for a versatile and programmable control of the air pump, accessible remotely through the WiFi-enabled user interface.

#### 2.1.4. Macrofluidic Prototyping for Plant-Based Scaffolds

The components in Figure 2 and Figure 3 afforded the assembly of a macrofluidic single-use bioreactor tailored for integration within conventional tissue culture laboratories. The primary objective of this design was to maintain a temperature of 37 °C, while continuously facilitating the delivery of CO_2_-enriched air from the incubator. The system incorporated meticulously designed ports to introduce flexible tubes. Each was affixed with 0.22-micron filters to enable a controlled and filtered air exchange secured by zip ties (Figure 2c). A port, employing a screw cap derived from a sterilized 15 mL tube, was allocated at the top for the introduction and extraction of media and scaffolds. The tubes and tube caps were autoclaved and assembled in a laminar flow hood. The inclusion of a conical bottom augmented the system’s functionality, allowing for versatile applications from 1 to 50 mL. In addition, we designed an area for a label under the conical bottom so that the user can write or put a printed sticker on the MSUB.

#### 2.1.5. Scaffold Sterilization

The scaffolds were placed in a glass jar with a lid and autoclaved for 30 min at 121 °C and 15 psi in a Tuttnauer 3870 ELV-D vertical autoclave (Tuttnauer USA Co., Hauppauge, NY, USA). Following sterilization by autoclave, the jar was opened in a laminar flow hood, and the rice puff scaffolds were subjected to UV for 30 min in the hood and then introduced into the multiwell plates or the MSUB through the tube cap using sterile tweezers.

### 2.2. Cell Culture

#### 2.2.1. Bovine MSC Immortalization

Bovine mesenchymal stem cells (bMSCs) were isolated, cultured, and characterized based on generally accepted criteria [69,70] as in [71]. The bMSC-SV40-hTERT cell line immortalization was achieved by introducing simian virus 40 large T antigen (SV40T, a kind gift from Prof. Sara Selig) by transient transfection, and the human telomerase gene hTERT [72] by retrovirus-mediated transduction. The cells were grown in low-glucose Dulbecco’s modified Eagle medium (Gibco, Carlsbad, CA, USA) containing 10% fetal bovine serum (Gibco, Carlsbad, CA, USA) and a penicillin–streptomycin mixture (3%) and cryopreserved in fetal bovine serum containing 10% DMSO. GFP-expressing cells were acquired using a pLKO_047 (Broad Institute) lentiviral vector and puromycin selection. Cell counts were determined with a cell counter (TC20 automated cell counter, (Bio Rad, Hercules, CA, USA).

#### 2.2.2. Expression of MSC Surface Markers

The expression of MSC positive surface proteins CD29, CD44, CD90, CD105, CD166, and negative marker krt19 [69,70] was determined using quantitative PCR. The procedure was carried out as described in [71]. In brief, RNA extraction from cells was performed using the PureLink RNA Midikit ((Invitrogen, Carlsbad, CA, USA) according to the manufacturer’s instructions. RNA was reverse-transcribed into cDNA using the High-Capacity cDNA Reverse Transcription Kit (Applied Biosystems, Waltham, MA, USA). Quantitative PCR was conducted using Fast SYBR Green Master Mix (Applied Biosystems) in an ABI Step-One Plus Real-time PCR system. To ensure validity, each sample was tested in triplicate (technical replicates). The relative mRNA fold change was calculated with the delta ct method, and the levels were normalized against those for bovine PSMB2.

#### 2.2.3. Seeding Cells on Rice Puff Scaffolds

Cells were seeded at 5000 (5K) to 500,000 (500K) cells per scaffold in 40 µL seeding volume. Following seeding, the scaffolds were placed in a 24-well plate and incubated for 45 min at 37 °C. Then, 1 mL of growth medium was added to each rice puff scaffold. The scaffolds were carefully transferred to a new 24-well plate using sterile tweezers three days later. Alternatively, in the MSUBs and flasks, we introduced ten puffed rice and seeded them in 400 µL NutriStem® MSC XF Medium (Satorius, Bohemia, NY, USA) containing 500K cells in three MSUB (as described in Figure 4a–e) and three 125 mL flasks with vent caps. The cells were maintained at 5% CO_2_ and at 37 °C in the MSUB. The air pump was programmed to provide intermittent CO_2_ enriched air from the incubator passed through the filter into the MSUB. Then, the medium in each well was replaced with 1 mL of fresh medium and 4 mL in the MSUB and flasks.

#### 2.2.4. Cellular Metabolic Activity

Metabolic activity was measured using “alamarBlue (Biorad)” at various time points, following the manufacturer’s guidelines. Scaffolds were incubated in 10% “alamarBlue” reagent in the growth medium for four hours at 37 °C, 5% CO_2_. Following incubation, 100 µL samples were transferred into a 96-well plate, and fluorescence was measured at 520 nm excitation and 590 emission wavelengths using a plate reader (BioTek Synergy HT Multi-Mode Microplate Reader; Agilent Technologies, Inc., Santa Clara, CA, USA). Results are reported as the fold change in dye reduction at each time point, normalized to the blank (rice without cells). For positive control, 5000 and 50,000 cells without scaffolds were seeded on 2D culture plates 24 h before plate reader analysis. The negative controls consisted of the growth medium, alamarBlue reagent, and puffed rice. Three biological replicates were performed for each experiment.

### 2.3. Microscopy

#### 2.3.1. Fluorescence Microscopy

Scaffolds were stained at room temperature using calcofluor-white (3 µL/mL and washed with PBS). Fluorescence imaging was performed using an Inverted Nikon ECLIPSE TI-DH fluorescent microscope (Melville, NY, USA). Confocal imaging was performed using a Leica SP8 confocal microscope (Teaneck, NJ, USA).

#### 2.3.2. Scanning Electron Microscopy

SEM images were acquired by a JSM-7800F (JEOL, Akishima, Japan) A field emission scanning electron microscope; puffed rice were dried with a K850 critical point dryer (Quorum Technologies Ltd., Lewes, UK) and coated with a Q150T ES Plus (Quorum Technologies Ltd.) at 18 mA for 90 s with 2 nm AuPd.

## 3. Results

### 3.1. Rice-Based Scaffold Microscopy, Water Absorption, and pH Assay

Puffed rice is already a dry porous material [73], and each scaffold was fitted in a 48-well plate (Figure 5a). We used a locally available brand without additives, imported in vacuum from Gujarat, India (Figure 5b), and autoclaved the scaffolds before use. Scaffold staining with calcofluor-white revealed its surface porous structure. The scaffold was tested for pH and liquid retention by conducting a water absorbance assay as shown in Figure 5c,d. Water absorption dynamics of the scaffold were determined by measuring the dry weight to establish a baseline (*n* = 5). Controlled water additions and subsequent wet-weight measurements allowed for the calculation of percent water absorption. Wet-weight measurements following each addition facilitated the calculation of percent water absorption using the formula [(wet weight − dry weight)/dry weight] × 100, which yielded insights into the hygroscopic properties of the scaffolds and the determination of the seeding volume of 40 µL per rice puff. The pH of DDW was compared to the pH of one rice puff in 1 mL of DDW (Eutech pH 700), and no significant alteration in pH was observed (DDW—6.21, DDW + rice puff—5.98). The porous scaffold is visible by fluorescent microscopy as seen in Figure 5e.

### 3.2. Scanning Electron Microscopy (SEM)

Surface and cross-section SEM images of dry puffed rice were taken. The porous structure of a rice puff was revealed in the cross section (Figure 6, top row) and on the surface (Figure 6 bottom row).

### 3.3. Cell Line Properties in 2D

The primary culture of bovine umbilical cord mesenchymal stem cells was extracted as described in [71]. These cells have a limited number of divisions, after which their division rate slows until they reach a state of cellular aging (senescence) and cease to divide. To obtain a stable line and high proliferation efficiency of cells, we disrupted the cell cycle control of the cells using SV40T, and the telomerase gene (hTERT) was added to allow cells to continue dividing without damaging genomic integrity. Our immortalized bMSC cell line presented a normal morphology (Figure 7a) and could be subcultured for approximately 150 days at a gradually decreasing rate with high survival rates (Figure 7b). The results indicated that SV40-hTERT cells, with or without the GFP reporter, had strong proliferation ability and no obvious senescence. The expression of the MSC-markers CD29, CD44, CD90, CD105, and CD166, and the absence of the epithelial marker KRT19, were similar in the transformed and primary cells (Figure 7c).

#### Proliferation Assay in Wells and in the MSUB

The cells were evaluated for proliferation in 3D on a rice puff scaffold in wells; the cells proliferated on the rice puff scaffold and reached maximum growth after 10 days (Figure 8a). When cultured in the MSUB and compared to flasks with vented caps, there was a triple fold change from day 2 to day 9 in fluorescence measurement of alamarBlue (Figure 8b). These findings demonstrated that the MSUB could support the proliferation of the SV40-hTERT-GFP-GFP cell line on multiple rice puff scaffolds cultivated in one chamber with overhead gas exchange. To investigate cell coverage, we took a seeded puffed rice after 12 days in the MSUB and then examined it under fluorescence and confocal microscopes (Figure 8c,d). This revealed cell penetration and growth throughout the entire scaffold. In addition, we used an SEM to evaluate rice puffs without cells and with cells after 12 days of cultivation (Figure 8e,f). There was a clear difference in puffed rice after cultivation when imaged by the SEM; the cells adhered to the surface and inside the scaffold and on its surface, they appeared as dried sheets, and as they were dehydrated in the preparation process, the extracellular matrix between the cells could also be seen as thin strings stretching between the cell membranes.

## 4. Discussion

In this study, we introduced a novel method to fabricate macrofluidic systems for cell cultivation, with a specific focus on bioreactor construction and the incorporation of food-grade rice-based scaffolds. The laser welding method we employed demonstrated a high efficiency and reliability, creating robust and leak-free seals within polyamide polyethylene (PAPE) films, all while minimizing heat-induced damage to the film. A noteworthy feature of PAPE is its clarity, which enables microscopy and spectral sensing applications. This versatility enhances its applicability across diverse cell cultivation methods and other bioprocesses. It is crucial to highlight that PAPE is extruded in high heat and pressure as a blow film. This process leads to sterility between the sheets, which the laser welding process preserves. Notably, PAPE is already designated and used globally as a food-grade packaging material [74]. To further enhance environmental sustainability, PET-based films are affordable, recyclable, and produced industrially globally [75,76]. Ensuring the durability of laser-welded connections in macrofluidic devices is crucial for large-scale industrial use. Scaling out, replicating processes on a smaller parallel level, is key to industrial production. Our materials are compatible with existing packaging machinery, supporting the mass introduction of scaffolds into multiple small-scale bioreactors. The robustness of the film we use is evident in tons of packed foods shipped and handled by machines and manually around the world. The material we used is fit for industrial use; according to the manufacturer, it has a tensile strength of 35 N/mm (4350 psi) in the machine direction (MD) and 30 N/mm (4350 psi) in the transverse direction (TD), along with respective tensile loads of 60 N and 50 N. Moreover, it exhibits elongation percentages of 580% (MD) and 640% (TD), while its impact resistance measures 1570 g and puncture energy stands at 400 N/mm. The integration of rice-based scaffolds into our system exemplifies the versatility of our macrofluidic fabrication method. Companies and researchers use different cells and different growth methods, and production pipelines require different solutions for different stages such as proliferation–expansion in suspension or differentiation–maturation on scaffolds; therefore, the versatility of the macrofluidic fabrication method is useful for startup companies that are in the development stage. Harmonizing with its sustainability and scalability objectives through the use of plant-based materials, an MSUB can be designed to fit scaffolds in different shapes and sizes with designated insertion ports that can be welded under sterile conditions after the scaffold insertion. We chose puffed rice as an example to show that our method can answer the unmet needs of a specific scaffold. However, the technology can be tailored to the needs of various scaffolds and processes. Under cultivation, over time, scaffolds may face shear forces that may cause damage to them; macrofluidics supports the design and implementation of a bioprocess with minimal shear forces to answer these needs. Aeration can be overhead, through the media in the culture area, or a separate gas exchange chamber; aeration is essential in order to supply oxygen to the cell culture [77]. Both the liquid and gas volume can be adjusted according to need by the design of the macrofluidic chamber and by adding media manually or automatically with a pump. Supervision over cultivation in real time can provide insights into scaled-out MSUBs in the bioprocess. For example, an MSUB that has a low seeding efficiency can be monitored in the first days of cultivation using microscopy or with a reagent such as alamarBlue and be replaced or reseeded; an MSUB that became contaminated can be sensed by sampling the media for turbidity manually or through the film with a light sensor [68] and discarded, potentially saving time and resources for a bioproduction facility that operates in a scale-out model. This supervision can be achieved by adding real-time sensors to each MSUB or by a monitoring station. Most cultured-meat cultivation is currently carried out in stirred-tank bioreactors without large 3D scaffolds [26]. Laser-printed macrofluidics can help address issues such as bioreactor clogging, inefficient mixing intensity, and the need for real-time monitoring; the spectral sensors we used in [68] may be used to sense the color change in phenol red during cultivation, mixing and clogging may be solved by a programmable pump, and for controlling the pump intensity by the design of the channels and chambers, we saw that designing a conical bottom and round bends in channels was useful in solving these problems. Verticality in our system is mandatory so that gasses go up and do not clog the channels or vessels in a way that may cause the fluidic process to not work; this is also true for perfusion bioreactors in general and is commonly solved with bubble traps. Additional fabrication may be useful for designing better ports that allow for the sterile introduction and removal of solids and liquids without the need for a laminar flow hood. The absence of an affordable commercial tissue bioreactor tailored for cultured-meat applications that allow for customization presents a significant economic challenge in this field. However, cost-effective materials such as PAPE film, as demonstrated in our study, can substantially reduce bioreactor costs by scaling out and not up. Using a sterile film during production eliminates the need for a separate sterilization step for the culture vessel, streamlining the process and contributing to cost efficiency. Potentially fitting for an aseptic VFFS (vertical form/fill/seal) [78] system supporting the scaffold insertion and sealing of the MSUB before downstream cultivation could also help streamlining the process. Moreover, the rapid prototyping capabilities afforded by this technique open the door to further research needed to address this critical hurdle in the cultured-meat production process.

## 5. Conclusions

Our results demonstrate the potential of laser welding of PAPE for the rapid fabrication of macrofluidic systems for cell cultivation. Combined with immortalized bovine stem cells cultured on plant-based-food-grade scaffolds, these findings support broader objectives of sustainable, cost-effective, and scalable cultured-meat production, presenting promising possibilities for the future of the cultured-meat industry and related fields.

## 6. Patents

WO2020105044—BIOLOGICAL FLUIDIC SYSTEM.

## Figures and Tables

**Figure 1 foods-13-01361-f001:**
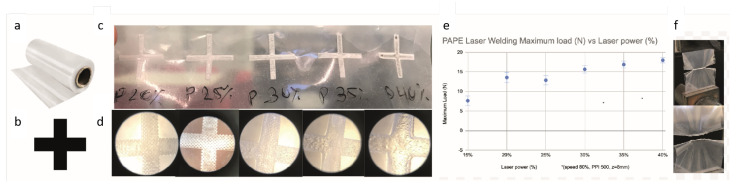
Calibration test for laser welding of PAPE bilayer. (**a**) PAPE roll as supplied by the factory. (**b**) Plus shape as designed using Adobe Illustrator. (**c**) Plus shape laser-welded on PAPE in different power settings. (**d**) Microscopic examination of the effect of different power settings on the welding quality and film heat deformation. (**e**) The maximum load was tested using an Instron testing machine. (**f**) Before and after maximum load while testing on an Instron testing machine.

**Figure 2 foods-13-01361-f002:**
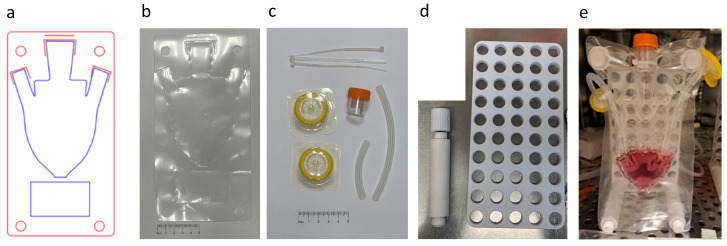
Macrofluidics. (**a**) Design of the macrofluidic device using Adobe Illustrator (blue: weld; red: cut). (**b**) Laser-printed single-use bioreactor. (**c**) Autoclaved flexible tubes and 0.22-micron membrane filters, tube cap, and zip ties. (**d**) Peg and backboard. (**e**) Assembled MSUB.

**Figure 3 foods-13-01361-f003:**
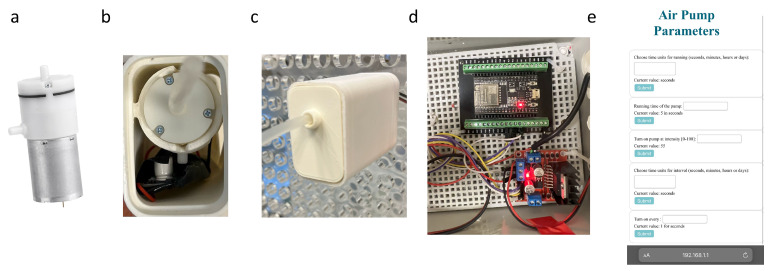
Air Pump for macrofluidic devices: (**a**) 12 V DC motor air pump; (**b**) air pump in 3D printed housing; (**c**) air pump attached to the backboard; (**e**) WiFi-accessible interface for control over the air pump; (**d**) ESP32 and L298N H-bridge control of the air pump.

**Figure 4 foods-13-01361-f004:**
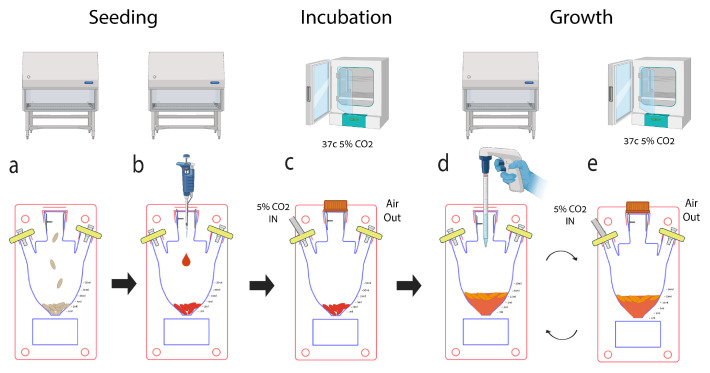
Process for seeding and cultivation of cells in an MSUB. (**a**) In a laminar flow hood, open the tube cap and insert dry scaffolds. (**b**) Using a micropipette seed the scaffolds with bMSCs with the appropriate volume to reach the absorption maximum (40 µL/rice puff). (**c**) In a CO_2_ incubator, program via the WiFi user interface, connect the MSUB to the air pump, and incubate for 1 h. (**d**) Disconnect from the air pump, and in a laminar flow hood, supplement the SUB with fresh media. (**e**) Reconnect the MSUB to the air pump in the incubator and start; add exchange media as needed.

**Figure 5 foods-13-01361-f005:**
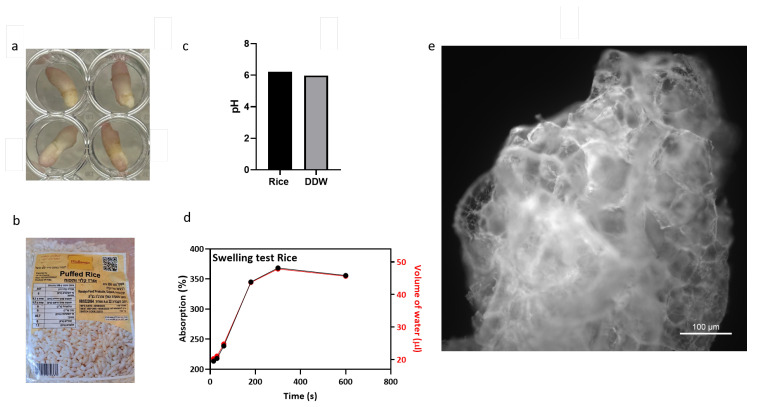
Rice puff properties. (**a**) Autoclaved and seeded puffed rice in a 48-well plate. (**b**) Vacuum-packaged puffed rice was acquired from a store. (**c**) pH assay. (**d**) Water absorbance assay. (**e**) A rice puff was stained with calcofluor-white and imaged using a fluorescence microscope (scale bar = 100 µm).

**Figure 6 foods-13-01361-f006:**
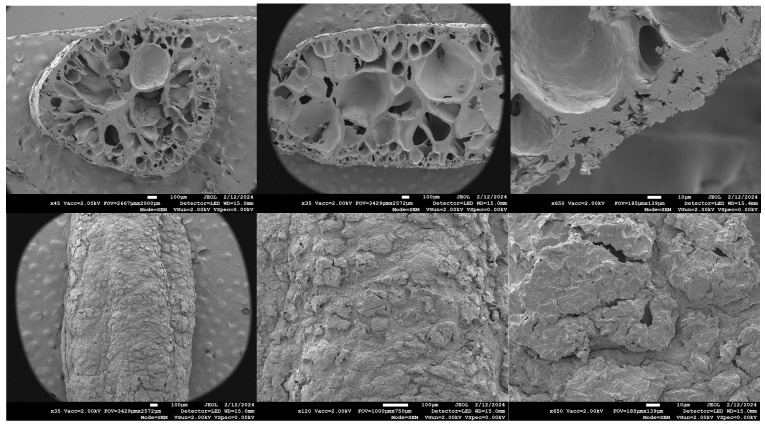
SEM images of puffed rice in the cross section and of the surface (top: cross section; bottom: surface).

**Figure 7 foods-13-01361-f007:**
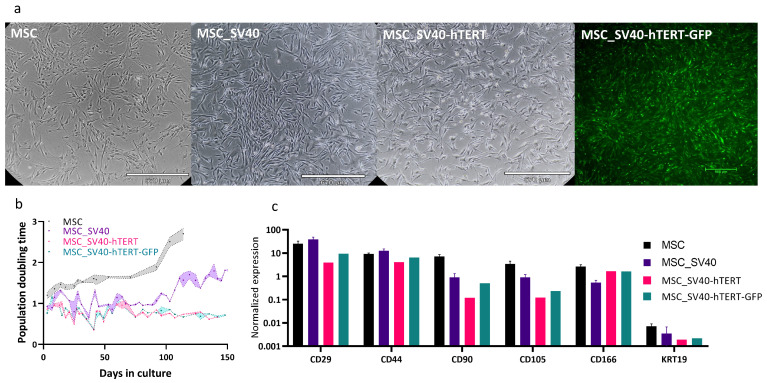
Characterization of immortalized MSC lines. (**a**) Morphology of primary MSCs (left), and MSCs immortalized by SV40 or SV40+hTERT show similar morphologies (light and fluorescent microscopy, scale bar 670 or 500 µm, respectively). (**b**) Population doubling time of the cell line over time. (**c**) The relative expression levels of positive (CD29-CD166) and negative (KRT19) stem cell markers were measured by RT-qPCR and normalized to the PSMB2 control gene. For MSC and $MSC_{SV40}$, data are the mean ± SEM; *n* = 3 for $MSC_{SV40-hTERT}$, and *n* = 1 for $MSC_{SV40-hTERT-GFP}$.

**Figure 8 foods-13-01361-f008:**
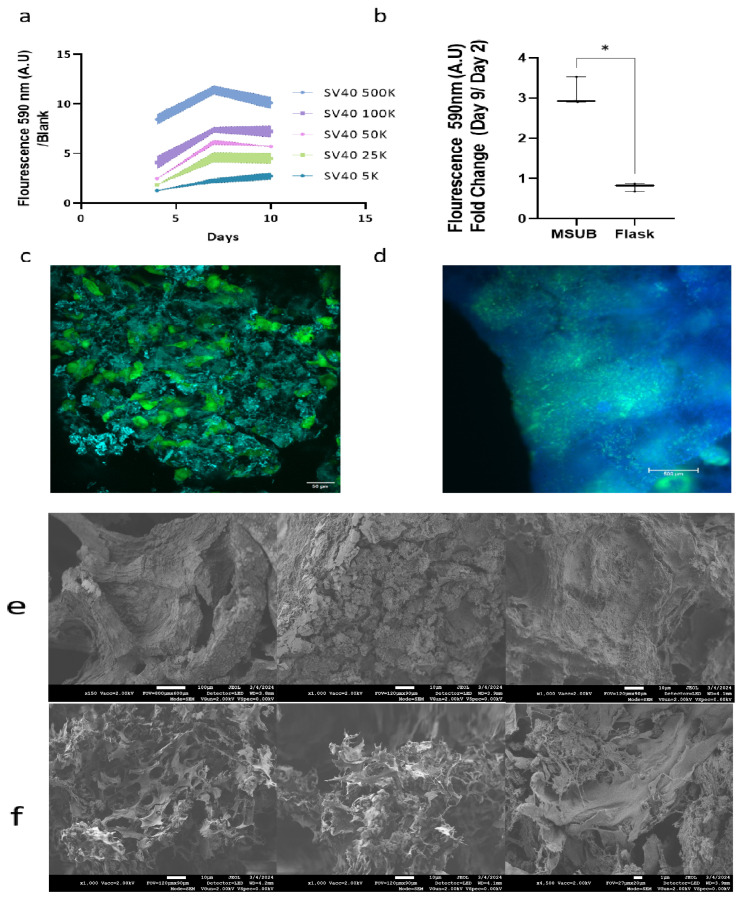
(**a**) Alamar Blue assay on puffed rice seeded with 5K–500K bMSCs normalized to the blank in multiwell plates. All data are represented as the mean ± SEM; *n* = 4. (**b**) Proliferation assay for 10 puffed rice samples seeded with 500K bMSCs in MSUBs vs flasks, *n* = 3. A one-tailed unpaired Mann–Whitney U test was used to calculate the statistical significance of the MSUB compared to that of the flasks. The asterisk indicates statistical significance. * *p* ≤ 0.05. (**c**) Representative images depicting rice puff scaffold (calcofluor-white (blue) seeded with MSC-SV40-hTERT-GFP (green) for 12 days). Images showing the scaffold surface were captured with a confocal microscope; scale bar: 50 µm. (**d**) Image captured with a fluorescent microscope, revealing the inner part of the scaffold. Scale bar = 500 µm. (**e**) SEM images of puffed rice after cultivation without cells. (**f**) SEM images of puffed rice after cultivation with cells.

## Data Availability

The original contributions presented in the study are included in the article; further inquiries can be directed to the corresponding authors.

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
