# Peer review of "Cultivation of Bovine Mesenchymal Stem Cells on Plant-Based Scaffolds in a Macrofluidic Single-Use Bioreactor for Cultured Meat"

_foods, 2024, doi:10.3390/foods13091361_

Round 1

Reviewer 1 Report

Comments and Suggestions for Authors

The research exhibits encouraging outcomes within the framework of laboratory-scale investigations. Nevertheless, How can one guarantee the enduring durability of laser-welded connections in macrofluidic devices and the potential for this method to be used for large-scale industrial production? 

The research emphasizes the benefits of the suggested macrofluidic device and plant-based scaffolds. However, a more comprehensive evaluation in relation to current bioreactor technologies and scaffold materials will enhance the persuasiveness of your approach in terms of its originality and practicality. This may encompass factors such as cost-effectiveness, scalability, and outcomes related to cell viability. 

It would be advantageous to include a section in the paper that explores present or prospective research endeavors focused on refining the manufacturing process, scaffold materials, or cell culture conditions with the objective of improving efficiency, yield, or product quality. This may encompass the investigation of various plant-based scaffolds or optimizing laser welding parameters to enhance the uniformity and durability of the macrofluidic devices.

Author Response

The research exhibits encouraging outcomes within the framework of laboratory-scale investigations.

We appreciate the reviewer's recognition of our study's importance. We have addressed the queries and revised the manuscript.

1) Nevertheless, How can one guarantee the enduring durability of laser-welded connections in macrofluidic devices and the potential for this method to be used for large-scale industrial production? 

We agree that ensuring the durability of laser-welded connections in macrofluidic devices is crucial for large-scale industrial use. Scaling out and replicating processes on a smaller parallel level is key to industrial production. Our materials are compatible with existing packaging machinery, supporting the mass introduction of scaffolds into multiple small-scale bioreactors. We are now addressing this in the discussion section, line 264-274. 

2) However, a more comprehensive evaluation in relation to current bioreactor technologies and scaffold materials will enhance the persuasiveness of your approach in terms of its originality and practicality. This may encompass factors such as cost-effectiveness, scalability, and outcomes related to cell viability. 

We thank the reviewer for the suggestion and added a short comparison of our device with current technologies in the introduction (lines 64-69). Current scaffold materials for bioreactors are costly and often not food-grade, while traditional bioreactors are designed for cell suspension cultures, not scaffold integration. Our method employs plant-based scaffolds, enhancing practicality and sustainability for large-scale cultured meat production.  We added a citation to this paper that most recently described the gaps and needs of the industry. 

3) It would be advantageous to include a section in the paper that explores present or prospective research endeavors focused on refining the manufacturing process, scaffold materials, or cell culture conditions with the objective of improving efficiency, yield, or product quality. This may encompass the investigation of various plant-based scaffolds or optimizing laser welding parameters to enhance the uniformity and durability of the macrofluidic devices.

We agree with the reviewer that there are objectives crucial for refining the manufacturing process, scaffolds have a critical influence on the bioprocess and in the context of cultured meat more factors arise such as regulation, cost and compatibility with the bioprocess itself.  We added considerations for scaffolds in the context of cultured meat. In an industrial food application scaffolds should be durable enough to go through the process, and be introduced to it with a minimal cost, both for the scaffold fabrication process and its sterilization. Finally, a scaffold that is compatible with cells and supports proliferation in a bioprocess should be regulated as a food.  Choosing scaffolds that are already regulated as foods solves one of these issues.  (lines 37-48).

References: 

Kulus, M., Jankowski, M., Kranc, W., Golkar Narenji, A., Farzaneh, M., DziÄ™giel, P., ... & Kempisty, B. (2023). Bioreactors, scaffolds and microcarriers and in vitro meat production—current obstacles and potential solutions. Frontiers in Nutrition, 10, 1225233.

Reviewer 2 Report

Comments and Suggestions for Authors

Opinion:

In this study, a laser welding method was developed to join thermoplastic films in a way that prevents contamination and leaks, affording effective macro-fluidics fabrication. This technique was tested using polyethylene (PET) films and a laser cutter operating with settings calibrated for the material to be welded or cut. The laser welding method was found to produce strong, leak-free seals in PET films with minimal heat-induced damage to the films. The ability to design fluidics, chambers and ports of various size. Using this method afforded the incorporation of plant-based scaffolds from food-grade plants (Rice). The laser welding method developed in this study provides a reliable, contamination-free method for the rapid fabrication of fluidic systems for cell cultivation. This technique has the potential to be compatible with a vast range of cell types and scaffolds and to be widely adopted in tissue engineering for regenerative medicine and food applications such as cultured meat. This article should be accepted by this journal with a little revision.

1. Figure 5c is not clear, please change a clear image.

2. Some captions in the figures have inconsistent font sizes like figure 8a and 8b, please make changes.

3. The SEM images of Figure 8e and 8f are not clear.

4. Some grammar errors in this manuscript should be revised.

5. Some writing errors in this manuscript should be revised like “37c”, “CO2”.

6. “(PWM))” in Line 103 needs to be corrected.

7. The first word in the “2.1.4. macrofluidic prototyping for plant based scaffolds” should be capitalized, check all your writing errors carefully.

8. How to autoclave the scaffolds rice puff scaffold before seeding cells? Please add more details.

Comments on the Quality of English Language

Some spelling and writing errors should be revised and avoided.

Author Response

We thank the reviewer for acknowledging the novelty and importance of our study. We have attempted to address the queries of the reviewer and reproof the text and hope that the revised manuscript is significantly improved.

  1. Figure 5c is not clear, please change a clear image.

We thank the reviewer for pointing this out; we have changed the picture to a black-and-white image of the dry puffed rice.  

  1. Some captions in the figures have inconsistent font sizes like figure 8a and 8b, please make changes.

We thank the reviewer for the comment, and fixed all font size inconsistencies.  

  1. The SEM images of Figure 8e and 8f are not clear.

We thank the reviewer for pointing this out, and changed the pictures to more focused ones. 

  1. Some grammar errors in this manuscript should be revised.

Fixed

  1. Some writing errors in this manuscript should be revised like “37c”, “CO2”.

Fixed

  1. “(PWM))” in Line 103 needs to be corrected.

Fixed

  1. The first word in the “2.1.4. macrofluidic prototyping for plant based scaffolds” should be capitalized, check all your writing errors carefully.

We thank the reviewer for pointing this out; we thoroughly revised all grammar and writing errors. 

  1. How to autoclave the scaffolds rice puff scaffold before seeding cells? Please add more details.

We appreciate the comment. We have now added a better description of this process to the materials and method section 2.1.5. 

We appreciate all the reviewers’ suggestions for improving the manuscript and believe that the careful revisions have significantly enhanced it.